# Chromosomal Location of *xa19*, a Broad-Spectrum Rice Bacterial Blight Resistant Gene from XM5, a Mutant Line from IR24

**DOI:** 10.3390/plants12030602

**Published:** 2023-01-29

**Authors:** Satoru Taura, Katsuyuki Ichitani

**Affiliations:** 1Institute of Gene Research, Kagoshima University, 1-21-24 Korimoto, Kagoshima 890-0065, Japan; 2The United Graduate School of Agricultural Sciences, Kagoshima University, Kagoshima 890-0065, Japan; 3Faculty of Agriculture, Kagoshima University, Kagoshima 890-0065, Japan

**Keywords:** mutation, *Oryza sativa*, resistance by a recessive gene, trisomic analysis, *Xanthomonas oryzae* pv. *oryzae*

## Abstract

Bacterial blight is an important rice disease caused by bacteria named *Xanthomonas oryzae* pv. *oryzae* (*Xoo*). XM5 is an *Xoo* resistant mutant line with the genetic background of IR24, an Indica *Xoo* susceptible cultivar, induced by a chemical mutagen N-methyl-N-nitrosourea (MNU). XM5 carries a recessive *Xoo* resistant gene, *xa19*. Trisomic analysis was conducted using the cross between XM5 and the trisomic series under the genetic background of IR24, showing that *xa19* was located on chromosome 7. The approximate chromosomal location was found using 37 surely resistant plants in the F_2_ population from XM5 × Kinmaze, which was susceptible to most Japanese *Xoo* races. The IAS44 line carries a Japonica cultivar Asominori chromosomal segment covering the *xa19* locus under the IR24 genetic background. Linkage analysis using the F_2_ population from the cross between XM5 and IAS44 revealed that *xa19* was located within the 0.8 cM region between RM8262 and RM6728. *xa19* is not allelic to the known *Xoo* resistant genes. However, its location suggests that it might be allelic to a lesion-mimic mutant gene *spl5*, some alleles of which are resistant to several *Xoo* races. Together with *xa20* and *xa42*, three *Xoo* resistant genes were induced from IR24 by MNU. The significance of chemical mutagen as a source of *Xoo* resistance was discussed.

## 1. Introduction

Rice (*Oryza sativa* L.), an extremely important crop worldwide, is also a model plant of monocots. Its genome was sequenced in 2005. In fact, rice was the first crop to be sequenced [1]. The rice genome has been an anchor for other grass species genomes based on their synteny relation [2,3]. Bacterial blight is an important rice disease caused by bacteria named *Xanthomonas oryzae* pv. *oryzae* (*Xoo*), ranked fourth among pathogenic bacteria in molecular plant pathology [4]. According to a comprehensive review [5], it is a Gram-negative and non-spore-forming bacteria, motile by a single polar flagellum. After it invades leaves through water pores or wounds, it spreads rapidly through the vascular system to the tissues around the infected site. Consequently, the bacterial blight symptoms appear. The *Xoo* genome has also been sequenced [6,7,8]. After these pioneering works, more than one hundred *Xoo* genomes were sequenced with the assistance of next-generation sequencers [9,10]. The two genomes have opened the door for greater molecular understanding of the classical gene for gene theory [11]. A recent review by Jiang et al. summarized the 11 pairs of cloned rice resistance genes against *Xoo* and the cognate *Xoo* avirulence genes [12].

Jiang et al. reported more than 40 bacterial blight resistant genes [12]. According to the Oryzabase gene list (https://shigen.nig.ac.jp/rice/oryzabase/locale/change?lang=en (accessed on 26 December 2022)), *XA1* (*XANTHOMONAS ORYZAE PV. ORYZAE RESISTANCE 1*) to *XA44* are listed. Furthermore, according to our survey, *xa-45(t)* [13], and *Xa46(t)* [14] were recently reported. We induced mutation to IR24 using a chemical mutagen N-methyl-N-nitrosourea (MNU), and developed three *Xoo* resistant mutant lines, XM5, XM6 and XM14 [15,16,17,18,19,20]. Although IR24 is susceptible to almost all *Xoo* strains, the three XM lines are resistant to all *Xoo* races tested [16,17,18,20]. XM6 carries one recessive mutant *Xoo* resistant gene *xa20* located on the long arm of chromosome 3 [17,20]. XM14 carries another mutant *Xoo* resistant gene *xa42* located on the short arm of chromosome 3 [18,19]. Their locations are not overlapping with the *Xoo* resistant genes reported earlier. Understanding mutations and their applications have paved the way for advances in the elucidation of the genetic, physiological and biochemical bases of rice traits [21]. Creating variation through mutation has therefore grown to be among the most important tools available to improve rice [21]. In this context, the mutation for *Xoo* resistance from a susceptible cultivar IR24, as well as development of near-isogenic lines [22], might be a basis for the genetic study of resistance against *Xoo* and new sources for breeding rice cultivars that are resistant against *Xoo*. Actually, XM5 is resistant to all the six Philippines *Xoo* races. It carries a mutant *Xoo* resistant gene *xa19* [16], of which mapping on the rice genome has not been undertaken.

Rice cultivars are classified into two varietal groups, Indica and Japonica [23]. This classification has been made according to morphological and physiological characters [24,25]. This classification corresponded to that by DNA sequences: Much DNA polymorphism was detected between Japonica and Indica cultivars, which contributed to DNA marker-based linkage analysis of many *Xoo* resistance genes including *xa20* and *xa42* in our group [18,19,20,26,27,28,29].

As reported from earlier studies [18,20], the F_2_ plants from the cross between Indica and Japonica cultivars carry a diverse genetic background showing many intermediate types of *Xoo* reaction. This phenomenon is partially explained by the QTLs affecting the degree of resistance [27,28,29]. Furthermore, the large variation in agronomic traits, such as the tiller number and plant height caused by the Indica–Japonica genetic difference, might increase the variation in lesion length [18]. To minimize such genetic noise, the *Xoo* resistance should be evaluated under the uniform genetic background. IAS lines, chromosomal segment substitution lines with a Japanese Japonica cultivar Asominori chromosomal segments under the IR24 genetic background [30], were adopted for precise linkage analysis of *xa20* and *xa42*.

This report describes the reaction of XM5 to Japanese *Xoo* races and the chromosomal location of *xa19*, using the combination of trisomic analysis, rough linkage analysis using the DNA polymorphism between Indica and Japonica, and precise linkage analysis using IAS lines.

## 2. Materials and Methods

### 2.1. Bacterial Races and Plant Materials

The *Xoo* bacterial races used for this study were five Japanese races: race I (strain T7174), race II (strain T7147), race III (strain T7133), race IV (strain H75373) and race V (strain H75304) and three Philippines races: race 1 (strain PXO 61), race 2 (strain PXO 86), and race 6 (strain PXO 99).

The International Rice Research Institute (IRRI) developed IR24, an Indica cultivar from the Philippines. It is susceptible to the above five Japanese races and six Philippine races: race 1, race 2, race 3 (strain PXO 79), race 4 (strain PXO 71), race 5 (strain PXO 112), and race 6 ([15,16,17,18,20] and Section 3). XM5 is a mutant line derived from IR24, induced by MNU. It is resistant to the six Philippines *Xoo* races above [15,16].

Khush et al. developed 12 primary trisomics of rice with an Indica cultivar IR36 genetic background [31]. However, IR36 carries a *Xoo* resistant gene, *Xa4*, which is not suitable for the genetic analysis of *Xoo* resistance. Therefore, trisomic series of IR24 were developed by backcrossing IR24 as the recurrent parent to original IR36 trisomic series [32]. In the present study, to ascertain the critical chromosome of *xa19*, trisomic analysis using the above trisomics with IR24 background was conducted. Six types of trisomics having IR24 background, Triplo 1, Triplo 6, Triplo7, Triplo 9, Triplo 10 and Triplo 11, were used. The digit after Triplo represents the extra chromosome number that each Triplo line carries [33]. These trisomics were susceptible to the six Philippine *Xoo* races. F_2_ populations from trisomic F_1_ plants derived from the crosses between six types of trisomics and XM5 were tested for the reaction against *Xoo* inoculation.

Kinmaze is a Japanese cultivar. Our preliminary DNA marker-based analysis suggests that it belongs to temperate Japonica. Kinmaze proved to be susceptible to Japanese *Xoo* races I, II and III [34]. Because the trisomic analysis suggested that *xa19* was located on chromosome 7 (see Section 3), the F_2_ population from the cross between Kinmaze and XM5 was subjected to rough linkage analysis of *xa19* with DNA markers on chromosome 7.

The IAS lines [30] were adopted for precise linkage analysis of *xa19*. The graphical genotypes of IAS lines are obtainable at http://www.shigen.nig.ac.jp/rice/oryzabase/strain/recombinant/genotypeIAS (accessed on 26 December 2022). Among them, IAS44 carries the Asominori chromosomal segment of chromosome 7, on which the initial linkage analysis (see Section 3) mapped the *xa19* locus, under the IR24 genetic background. Asominori is resistant to Japanese *Xoo* races I and V, although it is susceptible to races II, III and IV [35]. Our preliminary experiment demonstrated that IAS44 was susceptible to the five Japanese *Xoo* races above. The F_2_ population from the cross between IAS44 and XM5 was subjected to precise linkage analysis of *xa19*.

XM5, IR24, IAS44, and Kinmaze were tested for their reactions to the five Japanese *Xoo* races. They had been tested separately before and were tested under the same condition. Three plants from each line were inoculated with each *Xoo* race.

### 2.2. Genetic Analysis of xa19

To determine the critical chromosome of *xa19*, the trisomics above were crossed with XM5 as a pollen donor. The F_1_ plants were planted in the screenhouse in IRRI; trisomic F_1_ plants were selected using morphological characteristics. Trisomic F_1_ plants from the cross of Triplo 11 were identified by microscopic observations of pollen mother cells at the metaphase of meiotic division. Before transplanting the F_2_ populations, the F_2_ seedlings were separated roughly into trisomics and disomics by visual observation, and were transplanted separately in the screenhouse. They were tested for reaction to the Philippine *Xoo* races 1, 2 and 6 simultaneously, and were classified as either trisomics or disomics using the same method as that described for trisomic F_1_ plant selection. Before inoculation, tillers of the respective plants were divided into three with different colored vinyl ties: one color for each *Xoo* race. The strategy of ascertaining the critical chromosome followed an explanation by Okumoto and Tanisaka [36]: If *XAxa19* locus is located on the disomic chromosome, then the F_2_ segregation ratio of resistant type (*xa19xa19*): susceptible type (2*Xa19xa19*:1*Xa19Xa19*) is to be 1:3. However, when *xa19* locus is located on the trisomic chromosome, the genotype of trisomic F_1_ plant is *XA19XA19xa19*. The trisomic F_1_ plant produces male and female gametes in the ratio of 2*Xa19*:1*xa19*:1*Xa19Xa19*:2*Xa19xa19*. However, most of the male gametes with an extra chromosome cannot take part in fertilization because of their weak viability. Consequently, for disomic plants, the ratio of 1 [resistant type (*xa19xa19*)]:8 [susceptible type (*Xa19xa19*, *Xa19Xa19*)] is expected. Regarding trisomic plants, segregation ratios of two kinds are expected: one is 1 [resistant type (*xa19xa19xa19*)]:44 [susceptible type (*Xa19Xa19Xa19*, *Xa19Xa19xa19*, *Xa19xa19xa19*)] by chromatid segregation. The other is 1:35 by maximum equational segregation. Iwata (1997) presented a relevant review about the rationale of trisomic analysis and two types of trisomic chromosome segregation, chromatid segregation and maximum equational segregation [37]. Therefore, when *xa19* locus is located on the extra chromosome, the ratio of resistant plants is far smaller than 0.25. The goodness of fit of the observed segregation ratio to the expected ratio was examined by chi-square test.

In the F_2_ populations from the cross between Kinmaze and XM5, 299 F_2_ plants were inoculated with Japanese *Xoo* race III (strain T7133). Then 37 plants selected as surely resistant judged by visual observation were genotyped for DNA markers located on chromosome 7 (Table 1) [1,38,39,40,41].

In all, 210 F_2_ plants from the cross between IAS44 and XM5 were subjected to precise linkage analysis of *xa19* using DNA markers. They were inoculated with Japanese *Xoo* race II (strain T7147). A linkage map around *xa19* was constructed using Antmap [42]. The Kosambi’s function was adopted to calculate the map distance.

### 2.3. Evaluation of Xoo Resistance

The trisomic analysis of *xa19* was conducted in IRRI, Los Baños, Philippines. Experiments were conducted in a greenhouse and in a screenhouse. The greenhouse, overlaid with transparent glass, was used for nursing. The screenhouse is a plastic greenhouse in which concrete beds with paddy field soil were located. Both facilities were screened with fine net to prevent insect invasion. F_2_ seeds were sown in wooden seed boxes (52 × 47 × 10 cm). Three-week-old seedlings were transplanted in the concrete bed with plant spacing of 20 × 20 cm in a screenhouse.

The genetic analysis using the F_2_ populations from the cross between Kinmaze and XM5, and that between IAS44 and XM5 was conducted at the Experimental Farm of the Faculty of Agriculture, Kagoshima University, Kagoshima, Japan, following a process described by [19].

### 2.4. Inoculation of Xoo

Preparation of *Xoo* for inoculation followed a process described by [20]. Briefly, *Xoo* stocks had been stored in skim milk medium at −80 °C. Potato semi-synthetic agar medium [43] was used for culture and inoculum preparation. The inoculum was diluted with distilled water. The absorbance was adjusted to A = 0.05 (620 nm) using a spectrophotometer, which corresponds to concentration of about 10^8^ colony-forming units per milliliter.

The plants were inoculated using clipping method [44] when the most plants reached the booting stage. The reaction of the plants to *Xoo* was evaluated 18 days after inoculation, based on the lesion length and symptoms of the lesion. The progeny of the cross between XM5 and trisomics and that between XM5 and Kinmaze were scored as resistant (R) and susceptible (S), as judged from the combination of lesion length and symptoms of the lesion by visual observation. The progeny of the cross between XM5 and IAS44 were evaluated by measuring the lesion lengths of three leaves.

### 2.5. Molecular Techniques

Molecular techniques followed [18]. Briefly, DNA was extracted following [45] with some modifications. The PCR-mixture (5 μL) contained 10 ng of genomic DNA, 200 μM dNTPs, 0.2 μM of each primer, 0.25 U of Taq polymerase and 1× buffer containing MgCl_2_. PCR products were analyzed using electrophoresis in 10% (29:1) polyacrylamide gel, followed by ethidium bromide staining and ultraviolet light irradiation.

## 3. Results

### 3.1. The Reaction of XM5 to Japanese Xoo Races

XM5, IR24, Kinmaze and IAS44 (three plants per accession) were subjected to inoculation with five Japanese *Xoo* races. Table 2 shows that XM5 was resistant to all races tested, whereas IR24, Kinmaze and IAS44 were susceptible to all races tested.

### 3.2. Trisomic Analysis of xa19

F_2_ populations from trisomic F_1_ plants derived from the crosses between six types of trisomics and XM5 were tested for *Xoo* resistance (Table 3). Plants were insufficient for segregation analysis in some F_2_ populations. F_2_ populations from trisomic F_1_ plants obtained by crossing XM5 with Triplo 1, Triplo 9, Triplo 10, and Triplo 11 showed disomic segregation with a ratio of 1 resistant: 3 susceptible. The trisomic fraction of the F_2_ population from trisomic F_1_ plants derived from the cross between Triplo 6 and XM5 was found to be significantly different from the disomic ratio at the 1% level, although the segregation for resistance in disomic fraction of this F_2_ fitted the disomic segregation ratio. On the other hand, the segregations of the disomic fraction and trisomic fraction of the F_2_ population from trisomic F_1_ plants in the cross of Triplo 7 × XM5 were significantly different from the disomic ratio. They gave good fit to the theoretical trisomic segregation ratio [37]. These results indicated that *xa19* was located on chromosome 7.

### 3.3. Rough Mapping of xa19

After trisomic analysis, the F_2_ population from the cross between Kinmaze and XM5 was subjected to linkage analysis of *xa19* using 9 DNA markers, RM481, RM7479, RM5672, RM1253, RM3583, RM3859, RM214, RM500, and RM11 (Table 4). Our visual observation after inoculation of *Xoo* race III indicated that 299 F_2_ plants were classified into 73 resistant plants and 226 susceptible plants. As reported before [18,20], the cross between Indica and Japonica cultivars showed many intermediate types. Therefore, the 37 resistant plants judged by stringent observation were subjected to further analysis. Based on the assumption that all the 37 plants were homozygous for *xa19* gene, *xa19* should be linked closely with DNA markers at which all the plants were homozygous for the XM5 allele. The data of the F_2_ generation suggest that *xa19* is located between RM7479 and RM5672, as shown in Table 4. To narrow down the chromosomal location, the genotypes of the two markers, RM6574 and RM6728, were examined for F_2_ plants Nos. 1–6. Because the frequency of recombination between RM7479 and RM5672 was very low (approximately 4/64), we skipped the genotyping of RM6574 and RM6728 for the remaining plants, assuming that they were homozygous for the XM5 allele at the two DNA markers. The F_3_ generation of the F_2_ plants Nos. 1–6 (ca. 30 plants per F_2_ plant) were also subjected to a progeny test of *Xoo* resistance after inoculation with *Xoo* race III, confirming that they were fixed for the *xa19* gene. The genotyping of additional DNA markers, RM6574 and RM6728, revealed that *xa19* was located between the two DNA markers, RM6574 and RM5672. These results confirmed the result of trisomic analysis, and revealed the approximate location of *xa19*.

### 3.4. Linkage Analysis of xa19

The genotypes of IAS44 had been checked for the following 7 DNA markers on chromosome 7: C1057, X338, R1440, C451, R1245, R1789 and C213 (http://www.shigen.nig.ac.jp/rice/oryzabase/strain/recombinant/genotypeIAS (accessed on 26 December 2022). IAS44 is homozygous for the Asominori allele at the RFLP marker loci spanning C1057 and R1789 on chromosome 7. Only C213 on chromosome 7 was homozygous for IR24 allele. C1057 is originally a cDNA clone, of which the DDBJ accession number is D15667. The cDNA sequence information indicates that it is located on around 2,635 kb of chromosome 7 of the updated rice genome Os-Nipponbare-Reference-IRGSP-1.0 [46]. Also, R1789 was originally a cDNA clone, of which DDBJ accession number is D24363. It is located on around 26,530 kb. Subsequently, we checked the genotypes of IAS44 for the PCR-based DNA markers in our stock: IAS44 was a homozygote for the Asominori allele of DNA markers spanning RM481 and RM11 (Table 1). As described above, IAS44 was susceptible to the five Japanese *Xoo* races (Table 2). The F_2_ population from the cross between XM5 and IAS44 showed a bimodal distribution of lesion length after inoculation with *Xoo* race II (Figure 1). Using the lesion length of 6 cm as the dividing point, the 210 F_2_ plants were classified into 43 resistant plants and 167 susceptible plants. The segregation ratio 43:167 fitted 1:3, one-gene segregation (χ^2^ = 2.292, *p* = 0.13). Genotypes of all F_2_ plants were determined using five DNA markers: RM6574, RM21134, RM8262, RM6728 and RM5672. In Table 5, Plant Nos. 10 and 18–20 showed short lesion length, suggesting that they were resistant against *Xoo* race II, and thus homozygous for the *xa19* gene. Plant No. 11 showed long lesion length, and its genotypes of the DNA markers surrounding *xa19* were homozygous for XM5 or heterozygous. These results suggest that it is heterozygous at the *xa19* locus. The combination of the lesion length and genotypes of DNA markers of these plants suggested that *xa19* was located between RM8262 and RM6728. The 13 recombinants between RM6574 and RM5672 were subjected to the F_3_ progeny test for *Xoo* resistance using *Xoo* race II. The remaining plants were not subjected to the F_3_ progeny test because their genotypes of *xa19* could be inferred from the results of the F_2_ generation and these plants contributed little to narrowing down the *xa19* location. All the results confirmed that *xa19* was located between RM8262 and RM6728 (Table 5). Based on the inference that the genotype of the *xa19* locus was identical to RM6728 except F_2_ plant No. 11 (Table 5), the linkage map around *xa19* was constructed (Figure 2). The *xa19* gene was closely linked with RM6728 with a genetic distance of 0.3 cm calculated using Kosambi’s function. With the aid of the information related to the physical location of RFLP markers of the framework map by Harushima et al. [47] and PCR-based DNA markers in this study, the *xa19* gene was located on the short arm of chromosome 7, close to the centromere.

## 4. Discussion

The results of this study show that *xa19* was located on chromosome 7, flanked by the two DNA markers, RM8262 and RM6728, using the combination of trisomic analysis, primary linkage analysis using the DNA polymorphism between Indica and Japonica, and chromosomal segment substitution lines. Among 46 reported *Xoo* resistance genes, two genes were located on chromosome 7. Kumar et al. [48] reported that a dominant *Xoo* resistance gene *Xa33* is flanked by two SSR markers RMWR7.1 (4.080 Mb) and RMWR7.6 (4.129 Mb) on chromosome 7. These two markers are encompassed by two SSR markers RM21077 (4.060 Mb) and RM21122 (4.836 Mb) on chromosome 7. The primer information of RMWR7.1 and RMWR7.6 was not published, but chromosomal locations of these markers were available. Furthermore, primer information of RM21077 and RM21122 was published elsewhere [1]. They are located on 4028 kb and 4804 kb, respectively, on IRGSP 1.0, and on 4060 kb and 4836 kb on IRGSP Build 5.0 (https://rgp.dna.affrc.go.jp/E/IRGSP/Build5/build5.html (accessed on 26 December 2022)). Therefore, we infer that Kumar et al. [48] used IRGSP Build 5.0 as the reference genome [1]. The chromosomal region ranging from 4080 kb to 4129 kb on IRGSP Build 5.0 corresponds to that ranging from 4047 kb to 4100 kb on Os-Nipponbare-Reference-IRGSP-1.0. Therefore, *xa19* is not allelic to *Xa33* [48]. Vikal et al. reported that *xa8* is flanked between RM21044 and RM21045 on chromosome 7 [49]. Their location on Os-Nipponbare-Reference-IRGSP-1.0 is 3698 kb and 3707 kb. Therefore, *xa19* is not allelic to *xa8*.

Mizobuchi et al. reported their isolation of five lesion-mimic mutants that showed reduced symptoms after infection with the rice blast fungus from 13,000 M_2_ lines of rice, one mutant of which was allelic to the previously reported lesion-mimic gene *spl5-1*, and named *spl5-2* [50]. The homozygotes of *spl5-2* under the Taichung 65 background showed resistance to *Xoo* strains T7174 (race I) and T7133 (race III), although the original cultivar Taichung 65 was susceptible to them. Furthermore, the homozygotes of *spl5-1* under the Norin 8 background were found to be resistant to the two *Xoo* races, although the original cultivar Norin 8 was susceptible to them. Mizobuchi et al. also reported that other lesion-mimic mutants, *spl4*, *spl7*, *spl10*, *Spl12*, *spl13*, *spl14* and *Spl15* were also resistant to them, although their original lines were susceptible to them [50]. Among these lesion-mimic genes, *spl5* was reported to be located between morphological marker genes *g* and *Rc*, both of which had been reported to be located on chromosome 7 [51]. Chen et al. conducted fine mapping of *SPL5* and located this gene to the 15.1 kb region flanked by two DNA markers, InDel66 and CAPS7, both of which are located on 5564 and 5579 kb of the updated rice genome IRGSP 1.0 [52]. According to Oryzabase https://shigen.nig.ac.jp/rice/oryzabase/gene/detail/812 (accessed on 26 December 2022), *SPL5* is the same gene as Os07g0203700 in RAP-DB (http:// rapdb.dna.affrc.go.jp/ (accessed on 26 December 2022)) [53], and as LOC_Os07g10390 in the Rice Genome Annotation Project (rice.plantbiology.msu.edu/ (accessed on 26 December 2022)) [46]. It encodes the cleavage and polyadenylation specificity factor (CPSF). One base deletion in the recessive *spl5* allele (*spl5-1* in [50]) induced in Norin 8 by gamma ray irradiation [51] caused frame-shift mutation inducing a premature stop codon, leading to loss of about 41% amino acids (560 aa) in its C-terminus, including the whole CPSF_A domain [52]. It is likely that the loss of amino acids makes *spl5* protein lose its normal biological function. The candidate *xa19* chromosomal region contains *SPL5* gene. Therefore, some possibility exists that *xa19* is allelic to *SPL5*. Our earlier report did not mention the lesion-mimic morphology of XM5 [16]. Actually, our recent close observation revealed that XM5 sometimes shows slight brown spots in its leaves similarly to lesion-mimic mutants, although we did not identify the condition that makes XM5 lesion-mimic.

We have mapped three mutant *Xoo* resistant genes on the rice genome: they were located on the chromosomal region where a series of *Xoo* resistant genes from *Xa1* to *Xa46* were not located. Regarding the *xa20* and *xa42* genes, no homolog of the cloned *Xoo* resistant gene has been identified in the candidate chromosomal regions in the Rice Genome Annotation Project (http://rice.plantbiology.msu.edu/index.shtml (accessed on 26 December 2022)) [46] or The Rice Annotation Project Database (https://rapdb.dna.affrc.go.jp/ (accessed on 26 December 2022)) [19,20,53]. These results suggest that mutations induced by N-methyl-N-nitrosourea are a credible source of novel resistance mechanism(s) against *Xoo* [20]. In the case of *xa19*, this gene might be allelic to *spl5*, a lesion-mimic mutant gene resistant against *Xoo*. If so, *xa19* is distinct from the reported *spl5* gene in that brown spots on the leaves of XM5 are conditional, and not so severe as homozygotes of the reported *spl5* gene [50,52], suggesting that N-methyl- N-nitrosourea might also contribute to the resistance mechanism by creating a new allele. We are undertaking fine mapping of the *xa19* gene to clone this gene, using next generation DNA sequencer.

Most of the *Xoo* resistant genes found in extant resistant cultivars and wild relatives are dominant. Moreover, they follow the gene-for-gene theory, as suggested by Flor [11,22,54]. Of the four recessive *Xoo* resistant genes cloned earlier, three, *xa13* [55], *xa25* [56] and *xa41* [57], encode SWEET-type protein. Also, *xa5* [58] encodes the TFIIA transcription factor. Their corresponding cognate avirulence genes of *Xoo* all encode the TAL effector [12], suggesting that the recessive *Xoo* resistant genes also follow the gene-for-gene theory suggested by Flor [11]. However, all the three mutant resistant genes in our lab, *xa19*, *xa20* and *xa42*, are resistant against all the *Xoo* races tested: they can be designated as broad-resistant genes, which can contribute to a rice breeding program. The identification of these genes might also contribute to the new molecular mechanism of broad-spectrum *Xoo* resistance. According to the review by Sato et al., the mutagenic effects of nitrosamides including MNU are characterized by specific nucleotide substitution [59]; Particularly, the MNU induces predominantly a GC to AT transition, based on data obtained with human cells and *E. coli*. Such a substitution tendency was also reported in rice, and the mutation rate of an M_2_ mutant population was calculated as 7.4 × 10^−6^ per nucleotide, representing one mutation in every 135 kb genome sequence [60]. These facts suggest that a very small genetic change such as one amino acid substitution might produce a new *Xoo* resistant gene. Recent studies produced a *Xoo* resistant rice plant using genome-editing technique by inducing mutation to targeted genes through a reverse genetic approach [10,61,62,63]. However, forward genetic mutation approaches might create a new resistant gene at a locus on which resistant genes have never been identified. Zeng et al. [61] produced a CRISPR/Cas9-mediated mutation of *OsSWEET14* that confers resistance to *Xoo* without a yield penalty: a tradeoff between immunity and growth which is thought to occur as a result of prioritizing resource allocation to either of the two processes [64]. The combination of forward and reverse genetics on disease resistance is expected to contribute to the understanding and breeding of broad-spectrum-resistant rice cultivars without any yield penalty.

## 5. Conclusions

XM5 is an *Xoo* resistant mutant line with the genetic background of IR24, an Indica *Xoo* susceptible cultivar, induced by a chemical mutagen N-methyl-N-nitrosourea (MNU). XM5 carries a recessive *Xoo* resistant gene *xa19*. Genetic analysis showed that *xa19* was located within 0.8 cM region between RM8262 and RM6728 on chromosome 7. *xa19* is not allelic to the known *Xoo* resistant genes. However, its location suggests that it might be allelic to a lesion-mimic mutant gene *spl5*, some alleles of which are resistant to several *Xoo* races.

## Figures and Tables

**Figure 1 plants-12-00602-f001:**
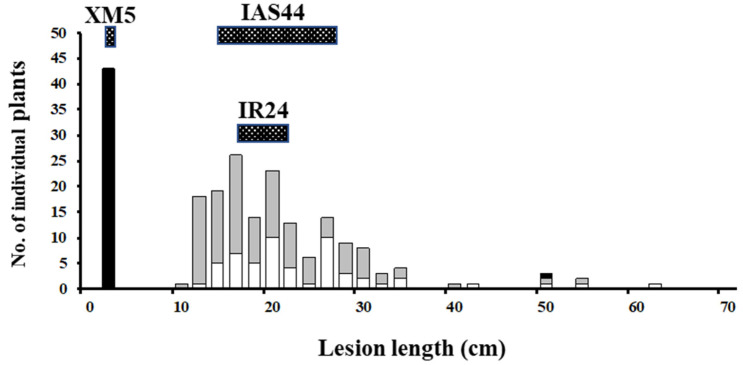
Frequency distribution of lesion length at 18 days after inoculation with *Xoo* Japanese race II (T7147) of the F_2_ population from the cross between XM5 and IAS44. The classified genotypes were assessed for RM6728 as noted: black, homozygous for XM5; gray, heterozygous; white, homozygous for IAS44. XM5 (10 plants), IAS44 (5 plants) and IR24 (10 plants) were planted as reference, and their lesion length ranges are shown as stippled boxes on the top.

**Figure 2 plants-12-00602-f002:**
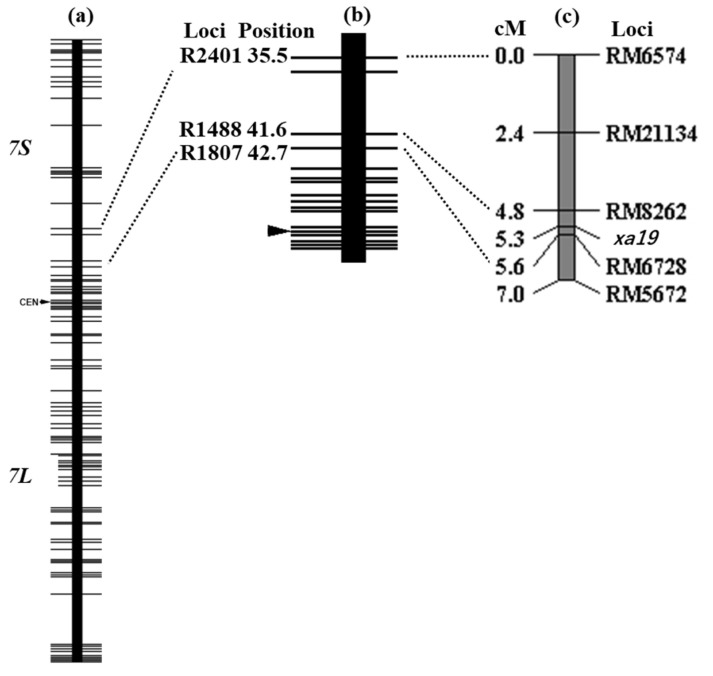
Linkage map of *xa19* and surrounding DNA markers. (**a**,**b**) RFLP framework maps of chromosome 7 modified from Harushima et al. [47]. (**c**) the linkage map of *xa19* gene constructed from F_2_ population from the cross between XM5 and IAS44 (*n* = 210). DNA markers located near each other on Os-Nipponbare-Reference-IRGSP-1.0 are connected by dotted lines.

**Table 1 plants-12-00602-t001:** Primer sequences of DNA markers for linkage analysis of *xa19* gene.

Marker Name	Direction	Primer Sequence	Location of Os-Nipponbare-Reference-IRGSP-1.0 of Chromosome 7		
			From (bp)	To (bp)	Source
RM481	F	TAGCTAGCCGATTGAATGGC	2,876,165	2,876,333	[38]
	R	CTCCACCTCCTATGTTGTTG			
RM7479	F	CCAGTTGCAACAAAGCTCTG	4,135,526	4,135,741	[39]
	R	TAGGAGCGTTTGTAGGAGCG			
RM6574	F	AACCTCGAATTCCTTGGGAG	4,682,829	4,682,937	[39]
	R	TTCGACTCCAAGGAGTGCTC			
RM21134	F	GCTTCCTCGAGGGATGGTACGG	4,947,993	4,948,107	[1]
	R	GCTGAATTCCAACTTTCCGAGACC			
RM8262	F	CATTAGCCGTGGTGTATTTG	5,299,401	5,299,600	[39]
	R	TTTCATCCCTAGTGCCAAC			
RM6728	F	GGGTATGTGTCGCTATTTTA	5,730,032	5,730,175	[39]
	R	GAAATCTGGAATTTTCCCTA			
RM5672	F	CACCCTACAAGGAAACAAGC	6,381,190	6,380,982	[39]
	R	TGCCCAATATAGAGGCAACC			
RM1253	F	CTGAACTTGCCTGAGAACTC	6,968,820	6,968,994	[39]
	R	GACGACCTCTCCATGCTCG			
RM3583	F	TACAATTTGGCGACCTCCTC	8,045,848	8,045,978	[39]
	R	GGATGCCATGTCATCATCTG			
RM3859	F	TTGCAGATCGGTTTCCACTG	8,878,209	8,878,399	[39]
	R	GGTCCTGGATTCATGGTGTC			
RM214	F	CTGATGATAGAAACCTCTTCTC	12,784,612	12,784,504	[40]
	R	AAGAACAGCTGACTTCACAA			
RM500	F	GAGCTTGCCAGAGTGGAAAG	15,911,539	15,911,794	[38]
	R	GTTACACCGAGAGCCAGCTC			
RM11	F	TCTCCTCTTCCCCC GATC	19,257,907	19,258,032	[41]
	R	ATAGCGGGCGAGGCTTAG			

**Table 2 plants-12-00602-t002:** Reaction in lesion length (cm) XM5, IR24 and IAS44 after inoculation with five Japanese races *Xanthomonas oryzae* pv. *oryzae*.

Rice Accession	Lesion Length ^1^ (cm)				
	Race I	Race II	Race III	Race IV	Race V
	T7174	T7147	T7133	H75373	H75304
XM5	1.8 ± 0.2	1.1 ± 0.6	2.2 ± 1.0	2.5 ± 0.5	1.1 ± 0.3
IR24	24.6 ± 5.7	30.2 ± 6.7	29.0 ± 3.5	20.2 ± 1.5	21.0 ± 4.8
Kinmaze	9.1 ± 3.1	13.9 ± 3.0	19.9 ± 4.0	5.7 ± 0.7	15.4 ± 2.3
IAS44	15.2 ± 3.9	19.7 ± 5.5	25.0 ± 1.2	11.4 ± 0.7	10.7 ± 1.1

^1^ Lesion length ± standard deviation at 18 days after inoculation.

**Table 3 plants-12-00602-t003:** Segregation for resistance to the bacterial blight pathogen in F_2_ populations from trisomic F_1_ plants derived from the cross between trisomics and XM5.

Cross	Fraction of Population				χ^2^			
		Reaction of F_2_ Plants ^1^			Expected Ratio			
		R	S	Total	1:3	1:8	1:44	1:35
Triplo 1 × XM5	Total	9	18	27	1			
Triplo 6 × XM5	2x	44	165	209	1.737	20.914 ** ^2^		
	2x+1	1	75	76	22.737 **		0.287	0.602
	Total	45	240	285	12.895 **			
Triplo 7 × XM5	2x	7	86	9	15.143 **	1.210		
	2x+1	0	49	49	16.333 **		1.114	1.400
	Total	7	135	142	30.507 **			
Triplo 9 × XM5	Total	54	178	232	0.368			
Triplo 10 × XM5	Total	16	79	95	0.372			
Triplo 11 × XM5	Total	46	189	235	3.689			

^1^ R and S, respectively, denote resistant and susceptible to Philippine *Xoo* race 1 (PXO 61), race 2 (PXO86), and race 6 (PXO 99). ^2^ ** means significant at 1% level.

**Table 4 plants-12-00602-t004:** Genotypes of DNA markers on chromosome 7 of 37 selected resistant plants against Japanese *Xoo* race III in the F_2_ population from the cross between XM5 and Kinmaze.

F_2_ Plant No. ^2^	Genotype of DNA Marker Loci ^1^										
	RM481	RM7479	RM6574	RM6728	RM5672	RM1253	RM3583	RM3859	RM214	RM500	RM11
1	K	H	H	X	X	X	X	X	X	X	X
2	H	H	H	X	X	X	X	X	X	X	X
3	H	H	H	X	X	X	X	X	X	X	X
4	X	X	X	X	X	X	X	X	X	X	H
5	X	X	X	X	X	H	H	H	H	H	H
6	X	X	X	X	H	H	H	H	H	H	H
7	X	X	-	-	X	X	X	X	X	X	X
8	X	X	-	-	ND	X	X	ND	X	X	X
9	X	X	-	-	ND	X	X	X	X	X	X
10	X	X	-	-	X	X	X	X	X	X	X
11	X	X	-	-	X	X	X	X	X	H	H
12	X	X	-	-	X	X	X	X	X	X	X
13	X	X	-	-	X	X	X	X	X	X	X
14	X	X	-	-	ND	X	X	X	X	X	H
15	X	X	-	-	X	X	X	X	X	X	X
16	X	X	-	-	X	X	X	X	X	X	X
17	H	X	-	-	X	X	X	X	X	X	X
18	X	X	-	-	ND	X	X	X	X	X	X
19	X	X	-	-	ND	X	X	X	X	X	H
20	X	X	-	-	X	X	X	X	X	X	X
21	H	X	-	-	X	X	X	X	X	X	X
22	X	X	-	-	X	X	X	X	X	X	X
23	X	X	-	-	X	X	X	X	X	X	X
24	X	X	-	-	X	X	X	X	X	X	H
25	X	X	-	-	X	X	X	X	X	X	X
26	X	X	-	-	X	X	X	X	X	X	X
27	X	X	-	-	X	X	X	X	X	X	H
28	X	X	-	-	X	X	X	X	X	X	X
29	X	X	-	-	X	X	X	X	X	X	H
30	ND	X	-	-	X	X	X	X	X	X	X
31	X	X	-	-	X	X	X	X	X	X	H
32	X	X	-	-	X	X	X	X	X	X	X
33	X	X	-	-	X	X	X	X	X	X	X
34	X	X	-	-	X	X	X	X	X	X	X
35	X	X	-	-	X	X	X	X	X	X	X
36	X	X	-	-	X	X	X	X	X	X	H
37	X	X	-	-	X	X	X	X	X	X	X

^1^ X, H, K, ND and -, respectively, denote homozygotes for XM5, heterozygotes, homozygotes for Kinmaze, genotypes not determined mainly due to error in molecular techniques, and genotypes not tested. ^2^ 1–6 were subjected to progeny test using F_3_ generation for inheritance of *Xoo* resistance and genotyping of additional DNA markers, RM6574 and RM6728.

**Table 5 plants-12-00602-t005:** Genotypes of informative recombinants for the DNA marker loci linked with *xa19* on chromosome 7 in the F_2_ population (XM5 × IAS44), and the gene segregation in the F_3_ generation.

F_2_ Plant No.	Lesion Length (cm)	Genotypes of the DNA Marker Loci ^1^					Segregation in F_3_ Plants ^2^		
		RM6574	RM21134	RM8262	RM6728	RM5672	R	S	Total
1	16.0	A	A	H	H	H	7	22	29
2	48.3	A	A	H	H	H	10	14	24
3	42.5	A	A	H	H	H	8	20	28
4	19.3	H	A	H	H	H	5	24	29
5	17.0	H	H	A	A	A	0	29	29
6	18.9	H	H	A	A	A	0	24	24
7	24.3	H	H	A	A	A	0	29	29
8	26.8	H	H	A	A	A	0	24	24
9	13.7	H	H	H	A	A	0	20	20
10	1.9	H	H	H	X	X	29	0	29
11	49.3	H	H	H	X	X	5	15	20
12	16.6	X	X	H	H	H	7	22	29
13	18.1	X	X	H	H	H	5	23	28
14	14.0	X	H	H	H	H	- ^3^	-	
15	29.0	X	H	H	H	H	-	-	
16	18.9	X	H	H	H	H	-	-	
17	13.0	X	H	H	H	H	-	-	
18	0.8	X	X	X	X	H	-	-	
19	0.3	X	X	X	X	H	-	-	
20	0.4	X	X	X	X	H	-	-	

^1^ X, H and A, respectively, denote homozygotes for XM5, heterozygotes, homozygotes for IAS44. ^2^ R and S, respectively, denote resistant and susceptible to *Xoo* race II and the gene segregation in the F_3_ generation. ^3^ indicates that F_3_ progeny were not tested for *Xoo* resistance.

## Data Availability

Data is contained within the article.

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
