# Peer review of "Chromosomal Location of xa19, a Broad-Spectrum Rice Bacterial Blight Resistant Gene from XM5, a Mutant Line from IR24"

_plants, 2023, doi:10.3390/plants12030602_

Round 1
Reviewer 1 Report
The present study has been done to map the xa19 allele. The XM5 line generated by chemical mutagen N-methyl-N-nitrosourea (MNU) in IR64 background was found resistant to bacterial blight contains xa19. Trisomic analysis was conducted using the cross between XM5 and trisomic series under the genetic background of IR24, which showed that xa19 was located on chromosome 7. Linkage analysis using the F2 population from the cross between XM5 and IAS44 revealed that xa19 was located within 0.8 cM region between RM8262 and RM6728. This is an excellent example which shows that, how chemical mutagenesis can be used to generate new alleles of economic importance. The English language of this manuscript needs improvement.
Comments:
1. Abstract: Line No.22 needs to be rewritten.
2. Material & Methods: Line No 74-75, not clear needs to be rewritten.
Line No. 79 replace “inconvenient” with “not suitable”.
Line No. 81-82 not clear needs to be rewritten.
Line No. 93-94 rewrite the sentence.
Line No. 101-102 not clear needs to be rewritten.
Line No. 133 not clear needs to be rewritten.
Line No.170 Not clear, rewrite.
3. Molecular techniques section (Line No. 180) needs to be written in detail so that this study can be reproduced by other researchers.
4. Conclusion: Line No. 410, needs to be rewritten.
Author Response
Reply to reviewer 1 is written in the attached file.

Reviewer 2 Report
Bacterial blight is an important rice disease caused by bacteria. XM5 is an Xoo resistant mutant line with the genetic background of IR24. This manuscript described the chromosomal location of xa19 by trisomic analysis and linkage mapping by F2 segregating populations. It was identified within 0.8 cM region between RM8262 and RM6728. The result had very large value for improving bacterial blight resistance for rice. However, I have two questions about this manuscript.
(1) For linkage mapping of xa19, the author should clearly explain that why they only perform progeny test for 6 lines in Table 4 and 13 lines in Table 5.
(2) which genotype was K in Table 4?
Author Response
Reply to reviewer 2 is written in the attached file.

Reviewer 3 Report
This study investigated the chromosomal location of xa19 on the XM5-mutant line derived from IR24 by mutagen N-methyl-N-nitrosourea. Authors efficiently utilized different types of genetic resources for mapping the targeted location. However, the writing and manuscript flow need improvement for easy understanding and interpretation of results.
Here are the modifications that need to be considered:
1. Abstract and material and methods are descriptive. For readers’ better understanding, authors could think to make it more specific.
The abstract should be enriched with key findings of the current study, rather authors’ previous reports. Similarly, in the case MM, background information can be shifted to the introduction or discussion section.
2. Line 259: X728: More information is needed about this maker with proper citation.
3. section 2.5: Authors should describe molecular techniques adopted with proper citation.
4. Table 4: column 1, row 1: what is K?
The conclusion from rough mapping is abrupt/sudden, need more data on the mapping results.
5. Line 237: RM6574 has data for very few plants that too only F3 generation. It creates confusion regarding results. Need more explanation for a better understanding.
6. Line 250: ND:not determined? Is it mean not tested/PCR not conducted? or not amplified in PCR?
7. Better to use a uniform coding system throughout the manuscript for genotype notation. Table 4, and table 5.
8. There are many typographical errors, Please carefully edit for errors like full stops, commas, and spaces.
Author Response
Reply to reviewer 3 is written in the attached file.

Round 2
Reviewer 1 Report
No Comments.
Reviewer 3 Report
The manuscript is now updated, and all corrections have been made.